# Spatial distribution, magnitude, and predictors of high fertility status among reproductive age women in Ethiopia: Further analysis of 2016 Ethiopia Demographic and Health Survey

**Desale Bihonegn Asmamaw**[1]*, **Wubshet Debebe Negash**[2], **Fantu Mamo Aragaw**[3], **Daniel Gashaneh Belay**[4], **Melaku Hunie Asratie**[5], **Abel Endawkie**[6], **Tadele Biresaw Belachew**[2]

1 Department of Reproductive Health, Institute of Public Health, College of Medicine and Health Sciences, University of Gondar, Gondar, Ethiopia, 2 Department of Health Systems and Policy, Institute of Public Health, College of Medicine and Health Sciences, University of Gondar, Gondar, Ethiopia, 3 Department of Epidemiology and Biostatistics, Institute of Public Health, College of Medicine and Health Sciences, University of Gondar, Gondar, Ethiopia, 4 Department of Human Anatomy, College of Medicine and Health Sciences, University of Gondar, Gondar, Ethiopia, 5 Department of Women's and Family Health, School of Midwifery, College of Medicine and Health Sciences, University of Gondar, Gondar, Ethiopia, 6 Department of Epidemiology and Biostatistics, School of Public Health, College of Medicine and Health Science, Wollo University, Dessie, Ethiopia

* desalebihonegn1988@gmail.com

## Abstract

### Background

Women's health and welfare, as well as the survival of their children, are adversely affected by high fertility rates in developing countries. The fertility rate in Ethiopia has been high for a long time, with some pockets still showing poor improvement. Thus, the current study is aimed to assess the spatial distribution and its predictors of high fertility status in Ethiopia.

### Methods

Secondary data analysis was used using the 2016 Ethiopian Demographic and Health Survey (EDHS). The Bernoulli model was used by applying Kulldorff methods using the SaTScan software to analyze the purely spatial clusters of high fertility status. ArcGIS version 10.8 was used to visualize the distribution of high fertility status across the country. Mixed-effect logistic regression analysis was also used to identify the predictors of high fertility.

### Result

High fertility among reproductive-age women had spatial variation across the country. In this study, a higher proportion of fertility occurred in Somali region, Southeastern part of Oromia region, and Northeastern part of SNNPR. About 45.33% (confidence interval: (44.32, 46.33) of reproductive-age women had high fertility. Education; no formal (aOR: 13.12, 95% CI: 9.27, 18.58) and primary (aOR: 5.51, 95% CI: 3.88, 7.79), religion; Muslim (aOR: 1.52, 95%

**Data Availability Statement:** Data for this study were sourced from Demographic and Health

surveys (DHS), which are freely available online at (https://dhsprogram.com).

**Funding:** The authors received no specific funding for this work.

**Competing interests:** The authors declare that they have no competing interests.

CI: 1.28, 1.81) and Protestant (aOR: 1.48, 95% CI: 1.23, 1.78), age at first birth (aOR: 2.94, 95% CI: 2.61, 3.31), age at first sex (aOR: 1.70, 95% CI: 1.49, 1.93), rural resident (aOR: 3.76, 95% CI: 2.85, 4.94) were predictors of high fertility in Ethiopia.

## Conclusion

The spatial pattern of high fertility status in Ethiopia is clustered. Hotspot areas of a problem were located in Somali, Central Afar, Northeastern part of SNNPR, and Southeastern part of Oromia region. Therefore, designing a hotspot area-based interventional plan could help to reduce high fertility. Moreover, much is needed to be done among rural residents, reducing early sexual initiations and early age at first birth, and enhancing women's education. All the concerned bodies including the kebele administration, religious leaders, and community leaders should be in a position to ensure the practicability of the legal age of marriage.

## Introduction

Women with high fertility are those who have at least five pregnancies at gestational ages greater than or equal to 20 weeks over their reproductive period, which is a major public health problem in developing countries, particularly in sub-Saharan Africa and Ethiopia [1,2]. It has negative health consequences for children and their mothers, slows economic growth, and exacerbates environmental problems [3].

Women with five or more pregnancies are at an increased risk for maternal death [4]. In addition, high-fertility countries have poor child survival rates. For instance, for every 100,000 births in African countries with high fertility, 640 women die during pregnancy and childbirth [5]. In Sub-Saharan Africa, population growth is outpacing economic growth. Many countries have fertility rates that are far higher than replacement levels [6,7]. African mothers replace themselves with nearly two daughters at a replacement fertility level of 2.1, which leads to rapid population growth [6].

Ethiopia has experienced high and persistent fertility rates for a long time. The TFR has declined from 5.5 children per woman in 2000 to 5.4 children per woman in 2005 to 4.8 children per woman in 2011 and to 4.6 children per woman in 2016 [8,9]. The fertility rate of Ethiopia is still high as compared to developed nations, even though the trend is declining [9]. The levels of maternal mortality and morbidity in Ethiopia are also among the highest in the world [9]. Short birth intervals, low maternal health service access, and high fertility seriously affect the health of mothers and children. Recent estimates showed that the country still experiences higher rates of maternal mortality, under-five mortality, and infant mortality of 412 deaths per 100,000, 67 deaths per 1000 live births, and 48 deaths per 1000 live births, respectively [9,10].

Of the Ethiopian population, more than 80% live in rural areas, whereas the high fertility and rapid population growth rate, especially in the rural areas, are unacceptably high, with a total fertility rate of above six children per woman [11]. In most Ethiopian rural communities, having a large family is still regarded as a source of pride and a divine gift [12]. Early age at first marriage, low socioeconomic status, culture, husband preferences, place of residence, region, education, and media exposure were significant factors leading to high fertility status [2,7]. High fertility status is also affected by the low level of health awareness and lack of access to modern contraceptives, especially in most of sub-Saharan Africa [2].

There is a highly skewed distribution of high fertility among reproductive-age women across socioeconomic, obstetric, and geographical lines. Ethiopia has not yet conducted spatial

analyses to identify areas with high fertility among reproductive-age women. Moreover, information regarding the magnitude and associated factors of high fertility among reproductive-age women in Ethiopia remains unclear. Understanding the level and geographical variation of high fertility status in Ethiopia can help health planners, programmers, partners in the health sector, and policymakers formulate appropriate strategies and interventions and provide quality reproductive health services to reduce high fertility. Hence, this study aimed to assess the spatial distribution of high fertility status and associated factors in Ethiopia.

## Methods

In Ethiopia, there are nine regions (Afar, Tigray, Amhara, Oromia, Somali, Southern Nations, Nationalities, and People's Region (SNNPR), Benishangul Gumuz, Gambella, and Harari) and two administrative cities (Addis Ababa and Dire Dawa) [13]. Based on Worldometer's analysis of the latest United Nations data, Ethiopia has 121,989,792 citizens as of Thursday, December 22, 2022 [14].

Secondary data analysis was done based on the 2016 Ethiopian Demographic and Health Survey (EDHS), which was a national representative sample conducted from January 18, to June 27, 2016. The EDHS 2016 was accessed from the DHS official database, www.measuredhs.com, after permission was secured through an online request by explaining the purpose of the study, which were used a cross-sectional study design and a two-stage stratified cluster sampling technique to select populations using the 2007 Population and Housing Census (PHC) as a sampling frame. Stratification was done by separating the nine regional states and the two city administrations of Ethiopia, into urban and rural areas [13].

A total of 645 Enumeration Areas (EAs) (202 in urban areas and 443 in rural areas) proportional to EA size were selected proportionally to the EA size in the first stage. In the second stage, 28 households in each cluster were selected with an equal probability of systematic selection [9]. For this study, the study population was all women of the reproductive age group from 25 to 49 years. this group of women was selected for this study by considering the fact that women in the study area who have given birth at early age have the possibility of giving birth five and more children before they celebrate their twenty fifth birthday [7]. We used individual datasets. A **total weighted sample of 9398 reproductive age women were included in the present study.** Additionally, latitude and longitude coordinates were taken from selected EAs (clusters). Full details of the EDHS sampling system were presented in the report [13,15].

### Study variables

The outcome variable was fertility status measured by the number of children ever born alive. It is categorized as high fertility when the number of children ever born alive is $\geq 5$ and low fertility when the number of children ever born alive is $< 5$. The cut-off point of 5 is taken because the medical and obstetric risk for women with the number of children ever born alive greater or equal to 5 is significantly higher compared with that of less than five [1,2,7,15].

Depending on different literature reviews, individual and community-level variables were included in the analysis. Women's education (no formal education, primary education, and secondary education and above), religion (Orthodox, Muslim, and Protestant), contraceptive methods use (not using any methods, short acting family planning, and long acting family planning), age at first sex ($< 18$ years, $\geq 18$ years), age at first birth ($< 18$ years, $\geq 18$ years), unmet need for family planning (unmet need, met need, and infecund/menopausal) were considered as individual-level variables. Media exposure; those who read newspapers, listened to the radio, or watched television at least once a week were coded yes and no otherwise [16]. The household wealth index was calculated using consumer goods like televisions, bicycles, and

cars. Materials used for the roof, floor, and toilet facilities were considered in calculating the household wealth index. To categorize individuals into wealth quintiles (poor, meddle, and rich), we used household asset data via principal component analysis (PCA) [17].

Of the community-level factors, distance to the health facilities (big problem, not big problems), and residence (rural, urban) were directly accessed from the EDHS data set. However, the aggregate community-level independent variables (community-level media exposure and community-level education) were constructed by aggregating individual-level characteristics at the community (cluster) level. They were categorized as high or low based on the distribution of the proportion values computed for each community after checking the distribution by using the histogram. The aggregate variable was not normally distributed, and the median value was used as a cut-off point for the categorization [18–20].

## Data management and analysis

For data analysis, we used STATA 14, ArcGIS 10.8, and SaTScan 9.6 software. For the analysis, sample weights were applied to adjust for the non-proportional sampling of strata and regions during the survey process and to restore representativeness. Text, figures, and tables were used to present descriptive statistics and summary statistics [13,20].

## Spatial analysis

**Spatial autocorrelation analysis.** The presence of spatial autocorrelation was identified using Moran's index (Moran's I). A Moran's I value close to -1 indicates that disease/events are dispersed, whereas a Moran's I value close to +1 indicates that they are clustered, and a Moran's I value of zero indicates that they are distributed randomly. There was a significant Moran's I ($p < 0.05$), indicating the presence of spatial autocorrelation and rejecting the null hypothesis (high fertility is randomly distributed). Hotspot analysis was conducted using the Getis-Ord Gi* statistic [13].

**Spatial scan statistical analysis.** Spatial scan statistics applied using Kulldorff's SaTScan software identified statistically significant primary (most likely) and secondary clusters of high fertility status. In SaTScan works, a window moves across the study areas and the window size needs to be fixed. As the outcome variable was Bernoulli distribution, Kulldorff's method was applieyd to use a Bernoulli model for a purely spatial analysis. In order to fit the Bernoulli model, respondents with high fertility were considered case, and those with low fertility were considered control. Using the default maximum spatial cluster size of 50% of the population as an upper limit, both small and large clusters were detected, and clusters with more than the maximum level were ignored. High fertility was considered in areas with a high Log Likelihood Ratio and significant p-value compared to areas outside the window.

**Multi-level analysis.** In the EDHS data, there was a hierarchical structure, which violates the independent observations and equal variance assumptions of a traditional logistic regression model. Therefore, women were nested within households, and households were nested within clusters. Within the cluster, they may have similar characteristics. Hence, multilevel binary logistic regression analysis must take into account the variability between clusters. Before adjusting for the variance through series of model development, we checked each variable at 0.2 p-values to include in the model. The final p-value remained <0.05 for the final model cut-point and AOR with 95% CI was also applied. Intra-class correlation coefficient (ICC), Median Odds Ratio (MOR), and Proportional Change in Variance (PCV) were computed to measure the variation between clusters. Taking clusters as a random variable, the MOR is defined as the median value of the odds ratio between the area at the highest risk and the area at the lowest risk area when randomly picking out two clusters. $\text{MOR} = e^{0.95\sqrt{VA}}$

Whereas, the ICC reveals the variation of high fertility between clusters is calculated as; $ICC = \frac{VA}{VA+3.29} * 100\%$. Moreover, the PCV reveals the variation in the high fertility among reproductive-age women explained by factors and calculated as; $PCV = \frac{Vnull-VA}{Vnull} * 100\%$ where; Vnull = variance of the initial model, and VA = area/cluster level variance [21–23]. We applied deviance (-2LLR) to compare models. The lower the deviance the more fitted the model.

## Ethical approval and consent to participate

Written informed consent was waived from the International Review Board of Demographic and Health Surveys (DHS) program data archivists after the consent manuscript was submitted to the DHS program/ICF to download the dataset for this study. The study is not an experimental study. All the methods were conducted according to the Helsinki Declarations. More details regarding DHS data and ethical standards are available online at http://www.dhsprogram.com.

## Results

### Socio-demographic related factors

A total weighted sample of 9398 women were included in this analysis. The median age of the study participants was 33 years (IQR: 28–39). About 31.43% of participants fell within the age category of 25–29 years, and 78.89% of the women were rural dwellers. Majority (65.52%) of the participants had no formal education. About 45.52% of participants were orthodox religious followers (Table 1).

### Obstetric related factors

About 68.17% and 42.22% of the women did not use any form of family planning methods and gave birth at home, respectively. More than two thirds (66.94) of the respondents had sex before 18 years old (Table 2).

### Regional proportion of high fertility status among reproductive age women in Ethiopia

The proportion of high fertility varies across the country. In this study, about 45.33% (confidence interval: (44.32, 46.33) of reproductive-ag women had high fertility. The highest and lowest proportion of high fertility were observed in the Somali region (66.54%) and Addis Ababa (4.86%), respectively (Fig 1).

### Spatial analysis of high fertility status

**Spatial autocorrelation and spatial analysis of high fertility status.** The spatial autocorrelation analysis revealed that the distribution of high fertility was non-random in Ethiopia, with a Global Moran's Index value of 0.99 (p<0.0001) (Fig 2). A higher proportion of fertility occurred in Somali region, Southeastern part of Oromia region and Northeastern part of SNNPR while low proportions of fertility were identified in the Addis Ababa, Harari, and Dire Dawa (Fig 3).

**Getis OrdGi statistical analysis of high fertility.** In the Getis OrdGi statistical analysis, significant hotspot areas (areas where fertility were high) were located in Somali, Central Afar, Northeastern part of SNNPR, and Southeastern part of Oromia region. Whereas the significant cold spot areas (areas with low fertility) were located in Addis Ababa, Dire Dawa, and Hareri (Fig 4).

**Table 1. Socio-demographic related factors of the participants of high fertility among reproductive age women in Ethiopia.**

| Variables | Frequency | Percentage |
|---|---|---|
| Region | | |
| Tigray | 619 | 6.59 |
| Afar | 71 | 0.76 |
| Amhara | 2318 | 24.66 |
| Oromia | 3399 | 36.16 |
| Somali | 271 | 2.88 |
| Benishangul Gumuz | 93 | 0.99 |
| SNNPRs | 2005 | 21.33 |
| Gambela | 25 | 0.27 |
| Harari | 22 | 0.24 |
| Addis Ababa | 523 | 5.56 |
| Dire Dawa | 52 | 0.56 |
| Residence | | |
| Rural | 7414 | 78.89 |
| Urban | 1984 | 21.11 |
| Age of the women | | |
| 25–29 | 2954 | 31.43 |
| 30–34 | 2333 | 24.82 |
| 35–39 | 1918 | 20.41 |
| 40–44 | 1252 | 13.32 |
| 45–49 | 942 | 10.02 |
| Education of the women | | |
| No formal education | 6158 | 65.52 |
| Primary education | 2133 | 22.70 |
| Secondary education and above | 1107 | 11.78 |
| Wealth index | | |
| Poor | 3355 | 35.69 |
| Middle | 1842 | 19.60 |
| Rich | 4202 | 44.71 |
| Media exposure to family planning messages | | |
| Yes | 3730 | 39.69 |
| No | 5668 | 60.31 |
| Distance to the health facilities | | |
| Big problem | 4847 | 51.57 |
| Not big problem | 4551 | 48.43 |
| Religion | | |
| Orthodox | 4278 | 45.52 |
| Muslim | 2966 | 31.56 |
| Protestant | 2155 | 22.93 |
| Community level education | | |
| High | 4435 | 47.19 |
| Low | 4963 | 52.81 |
| Community level media exposure to family planning messages | | |
| High | 4725 | 50.28 |
| Low | 4673 | 49.72 |

**Table 2. Obstetric-related factors of the participants of high fertility among reproductive age women in Ethiopia.**

| Variables | Frequency | Percentages |
|---|---|---|
| Place of delivery | | |
| Home | 3968 | 42.22 |
| Health institution | 5430 | 57.78 |
| Age at first sex | | |
| < 18 years | 6291 | 66.94 |
| ≥ 18 years | 3106 | 33.06 |
| Age at first birth | | |
| < 18 years | 3532 | 37.58 |
| ≤ 18 years | 5866 | 62.42 |
| Contraceptive methods use | | |
| Not using any methods | 6407 | 68.17 |
| Short acting family planning | 2102 | 22.36 |
| Long acting family planning | 890 | 9.47 |
| Unmet need for family planning | | |
| Unmet need | 2394 | 25.47 |
| Met need | 5217 | 55,51 |
| Infecund/menopausal | 1788 | 19.02 |

**Kriging interpolation.** In the Kriging interpolation; the predicted high fertility status were identified in Somali and South and eastern parts of the Oromia region whereas, the predicted low fertility status was identified in the Addis Ababa, Dire Dawa, and Hareri (Fig 5).

**Spatial Sa Tscan analysis of high fertility.** There were 565 significant clusters identified in the spatial Sa Tscan statistics, of which 86 were primary clusters (most likely). As a result of

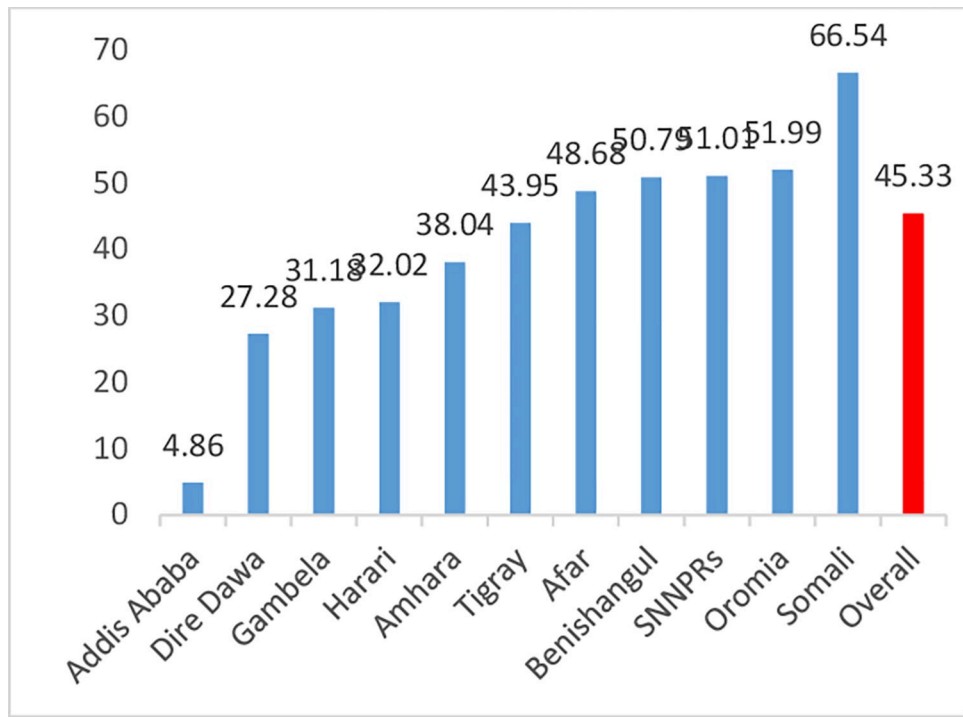

**Fig 1. Regional proportion of high fertility status among reproductive age women in Ethiopia.**

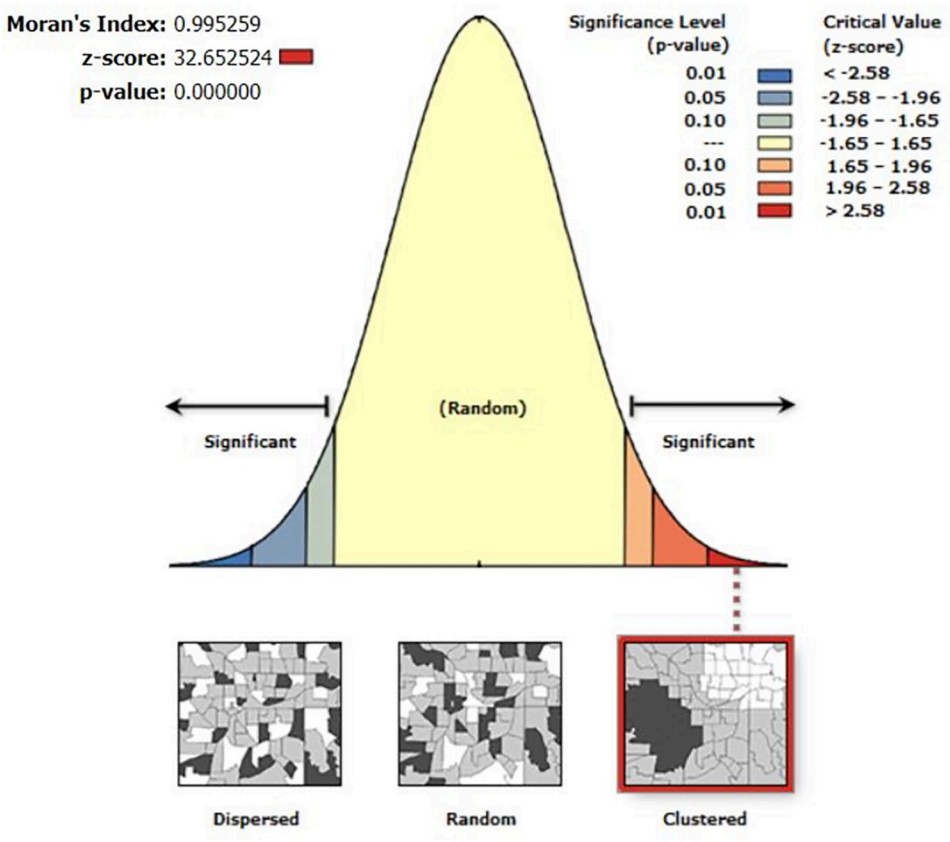

Given the z-score of 32.6525237968, there is a less than 1% likelihood that this clustered pattern could be the result of random chance.

**Fig 2. Spatial autocorrelation analysis of high fertility status among reproductive age women in Ethiopia.**

the survey, the primary clusters were found in the Somali, and eastern part of Oromia region. They were located at 5.330795 N, 41.837597 E of geographic location, with a radius of 440.63 km, with a Relative Risk (RR) of 1.6 and Log-Likelihood ratio (LLR) of 116.27, with p = 0.00001. According to the study, reproductive age women within the spatial window were 1.6 times more likely to have high fertility than reproductive age women outside it (Fig 6).

## Associated factors of TT immunization

**Random effect results.** The ICC value in the null model was 26.2% indicated that 26.2% of the total variability for high fertility as attributable to the between group variation while the remaining 73.8% was explained by the between individual variation. Besides, the MOR was 2.1 indicated that, if we randomly select two women from two different clusters, women at the cluster with a higher risk of high fertility had 2.1 times higher likelihood of high fertility compared with women at cluster with a lower risk of high fertility (Table 3).

**Fixed effect results.** In the multivariable multilevel logistic regression analysis; women's education, residence, religion, age at first sex, and age at first birth were predictors of high fertility in Ethiopia.

Accordingly, the odds of high fertility among women who had no formal education and had primary education were 13.12 (aOR: 13.12, 95% CI: 9.27, 18.58) and 5.51 (aOR: 5.51, 95%

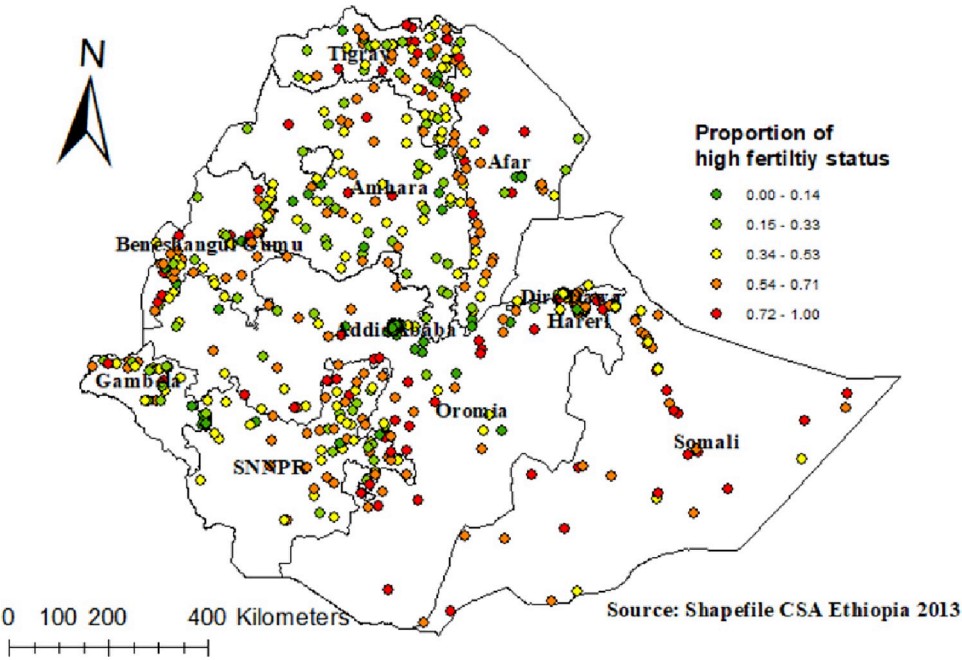

**Fig 3. Spatial distribution of high fertility status among reproductive age women in Ethiopia, Shape file source: Central Statistical Agency 2013, URL: https://africaopendata.org/dataset/ethiopia shape files. Map output: Own analysis using ArcGIS 10.8 software.**

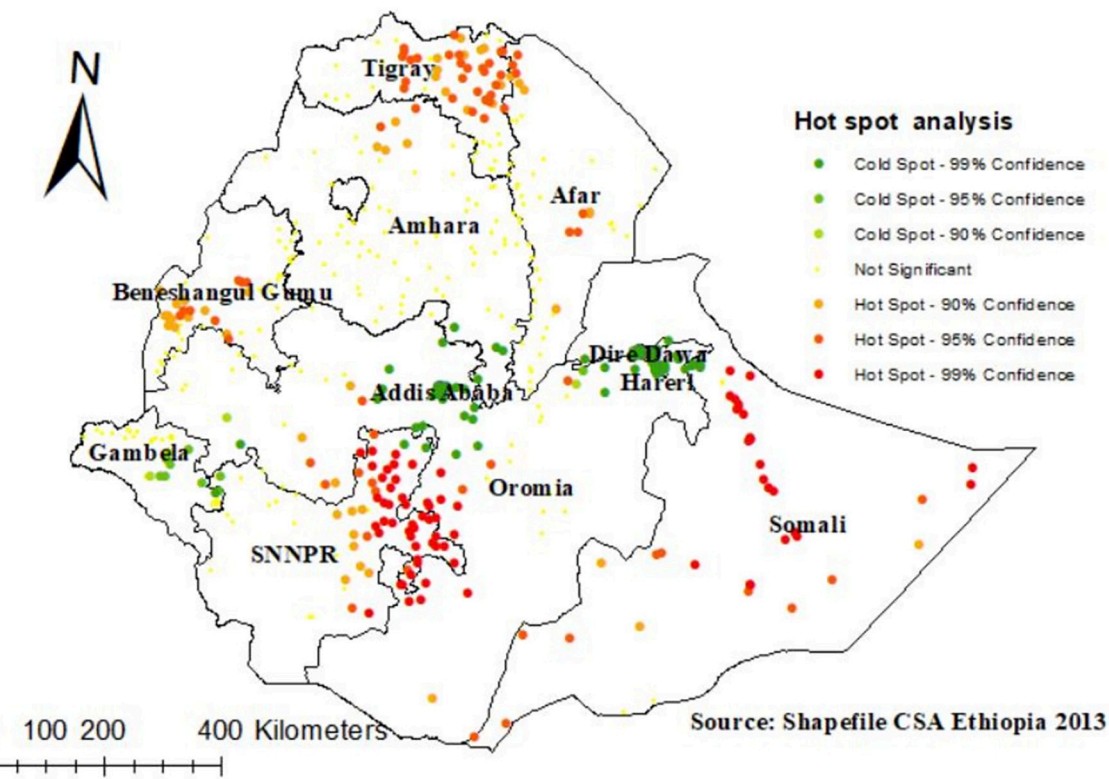

**Fig 4. Hot spot analysis of high fertility status among reproductive age women in Ethiopia, Shape file source: Central Statistical Agency 2013, URL: https://africaopendata.org/dataset/ethiopia shape files. Map output: Own analysis using ArcGIS 10.8 software.**

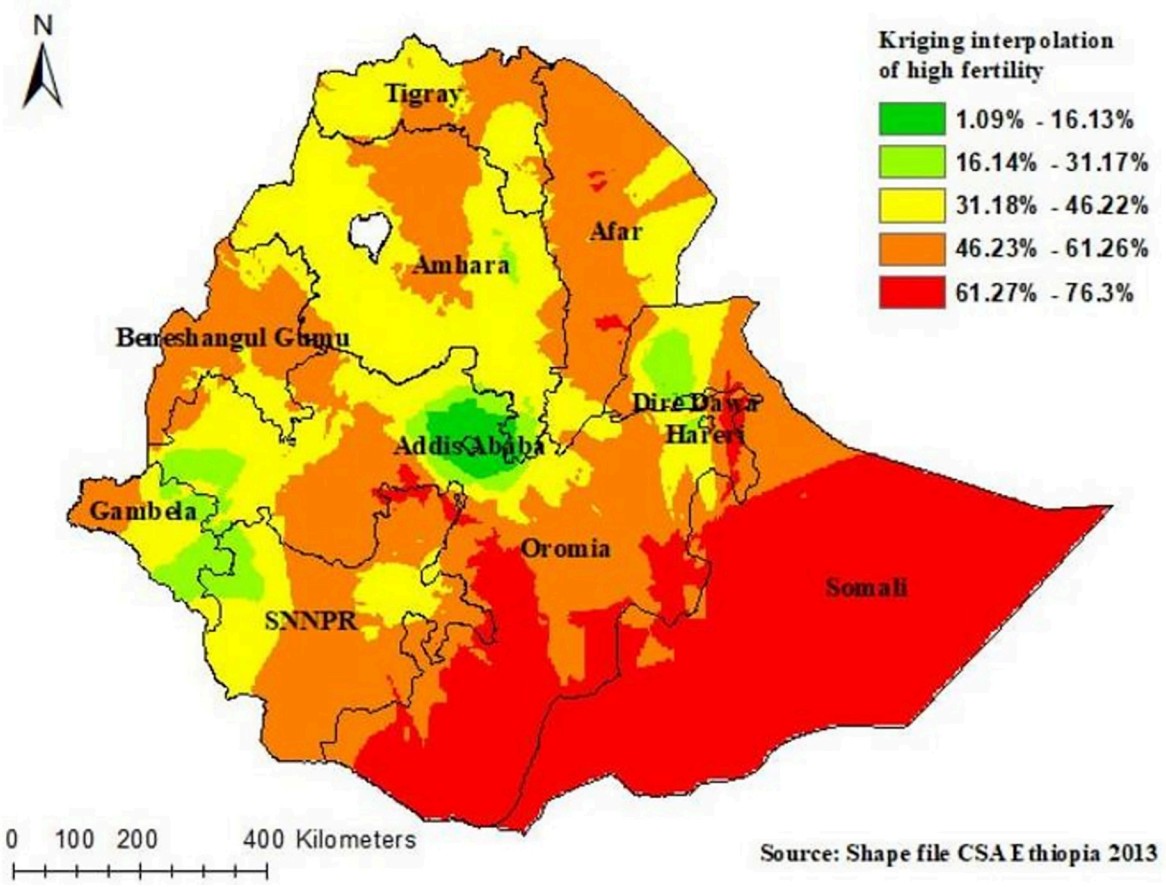

**Fig 5. Kriging interpolation of high fertility status among reproductive age women in Ethiopia, Shape file source: Central Statistical Agency 2013, URL: https://africaopendata.org/dataset/ethiopia shape files. Map output: Own analysis using ArcGIS 10.8 software.**

CI: 3.88, 7.79) times than those who had secondary education and above, respectively. Women who follow Muslim and protestant religion had 1.52 (aOR: 1.52, 95% CI: 1.28, 1.81) and 1.48 (aOR: 1.48, 95% CI: 1.23, 1.78) times more odds to have high fertility than those who follow orthodox religion. The odds of high fertility among women who gave birth first child before 18 years old (aOR: 2.94, 95% CI: 2.61, 3.31) were higher than those of women who gave birth the first child after 18 years old. The odds of high fertility among women who had their first sex before the age of 18 years old were 1.70 (aOR: 1.70, 95% CI: 1.49, 1.93) times higher than their counterparts. Moreover, women who lived in rural areas were 3.76 (aOR: 3.76, 95% CI: 2.85, 4.94) times more likely to have high fertility as compared with their counterparts (Table 3).

## Discussion

According to the present study, the spatial distribution of high fertility among reproductive-age women in Ethiopia was clustered. Significant hotspot areas with high fertility were identified in Somali, Central Afar, North-eastern part of SNNPR, and South-eastern part of Oromia region. Whereas the significant cold spot areas with low fertility were identified in Addis Ababa, Dire Dawa, and Harari. The possible justification for this difference could be due to the different socio-economic and obstetric-related factors of the study participants. For instance, the majority of the respondents in Addis Ababa, Dire Dawa, and Harari were educated, accessible to reproductive health services like family planning, and had a good awareness of

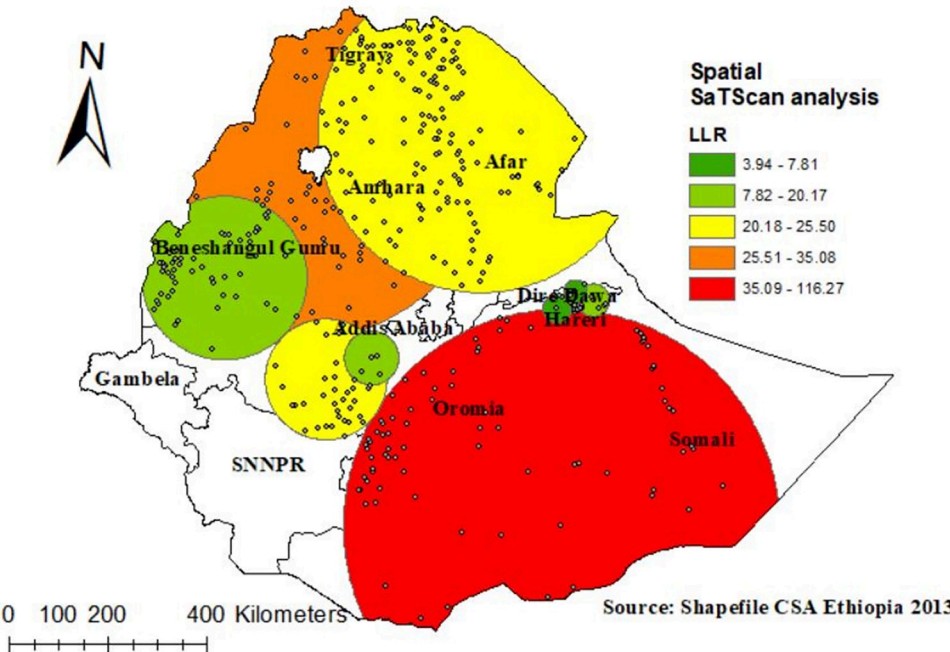

**Fig 6. SaTScan analysis of high fertility status among reproductive age women in Ethiopia, Shape file source: Central Statistical Agency 2013, URL: https://africaopendata.org/dataset/ethiopia shape files. Map output: Own analysis using ArcGIS 10.8 software and SaTScan 9.6 software.**

reproductive health services [24,25]. In addition, most participants who lived in these regions were exposed to the media. Which in turn enhances the reproductive age women's awareness on the effects of high fertility.

In the current study, residency was significantly associated with high fertility status. Women who reside in rural area have more children as compared to their counterparts. This is supported by studies done in Butajira [26], Ghana [27], and Nepal [28]. The possible justification might be that those who live in the rural area no longer stay in school, thereby married at school age [29]. Moreover, the demand for human power in agrarian living condition and the notion of considering the family with a large number of children as a blessed family in the rural area may drive couples to have an increased number of children. Moreover, access to media, general health knowledge, and better health service information, mainly characterizes urban women [30,31].

Another factor that affect the high fertility status found in our study was religion. Women who follow Muslim and protestant religions were more likely to have high fertility as compared with those who follow the orthodox religion. The possible reason might be the high proportion of Muslims (Afar, Somali, and Harar) and regions with a high proportion of Ethiopian Orthodox Christians (Addis Ababa, Amhara, and Tigray); fertility is much higher in developing regions like Somali; and most of the women in these regions have no formal education [9].

The odds of high fertility was higher among women with no formal education as compared to those with secondary education and above. This is in line with studies conducted in the Sidama region, Ethiopia [15], Nigeria [32], and Kenya [33]. The possible reason could be that women that are more educated may learn different ideas of desired family size through school, community, and exposure to global communication networks and know more about the risks of early birth and a short birth interval [34]. Moreover, educated women also know more about child health. Education generally results in an improvement in the status of women in society through a better understanding of health issues and employment status [33].

**Table 3. Multi-level mixed-effect logistic regression analysis of predictors of high fertility reproductive age women in Ethiopia.**

| Variables | Null model | Model I | Model II | Model III |
|---|---|---|---|---|
| Education of the mother | | | | |
| No formal education | | 17. 49 (12.41, 24.67) | | 13.12 (9.27, 18.58) |
| Primary education | | 6.83(4.84, 9.65) | | 5.51 (3.88, 7.79) |
| Secondary education and above | | 1 | | 1 |
| Religion of the respondents | | | | |
| Muslim | | 1.61 (1.35, 1.92) | | 1.52(1.28, 1.81) |
| Protestant | | 1.63 (1.35, 1.97) | | 1.48(1.23, 1.78) |
| Orthodox | | 1 | | 1 |
| Wealth index | | | | |
| Poor | | 0.92 (0.79, 1.07) | | 0.84 (0.73, 1.05) |
| Middle | | 0.94 (0.81, 1.10) | | 0.87 (0.76, 1.09) |
| Rich | | 1 | | 1 |
| Unmet need for family planning | | | | |
| Unmet need | | 1.55 (1.32, 1.81) | | 1.23 (0.97, 1;75) |
| Met need | | 0.95 (0.81, 1.12) | | 1.51 (0.75, 1.04) |
| Fecund/menopausal | | 1 | | 1 |
| Age at firs birth | | | | |
| < 18 years | | 2.89 (2.57, 3.25) | | 2.94 (2.61, 3.31) |
| ≥ 18 years | | 1 | | 1 |
| Contraceptive methods use | | | | |
| Not using any methods | | 1.19 ( | | 1.13 (0.92, 1.38) |
| Short acting family planning | | 1.26 (0.97, 1.46) | | 1.22 (0.98, 1.49) |
| Long acting family planning | | 1.23 (1.02 1.54) | | 1.19 (0.96, 1.48) |
| Age at first sex | | | | |
| < 18 years | | 1.75 (1.53, 1.98) | | 1.70 (1.49, 1.93) |
| ≥ 18 years | | 1 | | 1 |
| Media exposure to family planning message | | | | |
| Yes | | 1 | | 1 |
| No | | 1.13(0.99, 1.28) | | 0.98 (0.87, 1.12) |
| Residence | | | | |
| Rural | | | 6.95 (5.39, 8.94) | 3.76 (2.85, 4.94) |
| Urban | | | 1 | 1 |
| Distance to the health facilities | | | | |
| Big problem | | | 1.15 (1.03, 1.28) | 1.19 (1.06, 1.34) |
| Not a big problem | | | 1 | 1 |
| Community media exposure | | | | |
| Low | | | 1.57 (0.94, 2.63) | 1.58 (0.94, 2.66) |
| High | | | 1 | 1 |
| Community level education | | | | |
| Low | | | 0.91 (0.54, 1.52) | 0.82 (0.49, 1.40) |
| High | | | 1 | 1 |
| Random effect result | | | | |
| ICC | 26.2 | 14.8 | 12.6 | 12.3 |
| Variance | 61.3 | 57.2 | 47.6 | 46.1 |
| MOR | 2.1 | 2.05 | 1.9 | 1.8 |
| PCV (%) | | 6.7 | 22.4 | 24.8 |
| Model fitness | | | | |

*(Continued)*

**Table 3.** (Continued)

| Variables | Null model | Model I | Model II | Model III |
|---|---|---|---|---|
| LL | -5828.8 | -5034.6 | -5648.3 | -4958.4 |
| Deviance | 11657.6 | 10069.2 | 11296.6 | 9916.8 |

The likelihood of high fertility among women who had their first sex before 18 years old was higher than among those who had their first sex after 18 years. This could be because sex at an early age is a leading obstetric characteristic that indicates the exposure of women to teenage pregnancy [35]. This study also showed that high fertility was also higher among women who had their first birth before the age of 18 as compared to their counterparts. This is similar to the studies conducted in the Gedeo Zone, Ethiopia [7], and Nigeria [32]. This could be because women who started birth before 18 years, the period of fertility is longer, and they have many ever-born children. Usually, those women who started giving birth before the age of 18 resided in rural areas and had no formal education. As a result, they had poor knowledge of and access to reproductive health services like family planning methods [36].

The main strength of the current study was the use of weighted data that were nationally representative. As a result, the results of the current study can be applied nationally. In addition, we also identify similar and statistically significant areas with a high cluster of high fertility using both ArcGIS and Sat Scan statistical tests. As the study is cross-sectional in nature, a cause-and-effect relationship cannot be established. Moreover, due to the data being quantitative in nature, there are some hidden factors that need to be addressed by a qualitative study, and there might be biases like age at first birth and age at first sex.

## Conclusion

The spatial pattern of high fertility in Ethiopia is clustered. The significant hotspot areas (areas where fertility were high) were identified in Somali, Central Afar, the northeastern part of SNNPR, and the southeastern part of the Oromia region. Women's education, residence, religion, age at first sex, and age at first birth were predictors of high fertility in Ethiopia. Therefore, designing a hotspot area-based interventional plan could help reduce high fertility. Moreover, much needs to be done among rural residents, including reducing early sexual initiations and the early age at first birth and enhancing women's education. All the concerned bodies, including the kebele administration, religious leaders, and community leaders, should be in a position to ensure the practicability of the legal age of marriage.

## Acknowledgments

We are grateful to the DHS programs, for the permission to use all the relevant DHS data for this study.

## Author Contributions

**Conceptualization:** Desale Bihonegn Asmamaw.

**Data curation:** Desale Bihonegn Asmamaw, Wubshet Debebe Negash, Daniel Gashaneh Belay, Abel Endawkie, Tadele Biresaw Belachew.

**Formal analysis:** Desale Bihonegn Asmamaw, Wubshet Debebe Negash, Abel Endawkie.

**Investigation:** Desale Bihonegn Asmamaw.

**Methodology:** Fantu Mamo Aragaw.

**Software:** Desale Bihonegn Asmamaw, Fantu Mamo Aragaw, Melaku Hunie Asratie.

**Validation:** Desale Bihonegn Asmamaw.

**Visualization:** Desale Bihonegn Asmamaw.

**Writing – original draft:** Desale Bihonegn Asmamaw.

**Writing – review & editing:** Desale Bihonegn Asmamaw, Wubshet Debebe Negash, Melaku Hunie Asratie, Tadele Biresaw Belachew.

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
