## [Decision Letter · Decision Letter 0]

13 Jun 2023

PONE-D-23-07027Spatial distribution, magnitude, and predictors of high fertility status among reproductive age women in Ethiopia: Further analysis of Ethiopia Demography and Health SurveyPLOS ONE

Dear Dr. Asmamaw,

Thank you for submitting your manuscript to PLOS ONE. After careful consideration, we feel that it has merit but does not fully meet PLOS ONE’s publication criteria as it currently stands. Therefore, we invite you to submit a revised version of the manuscript that addresses the points raised during the review process.

We look forward to receiving your revised manuscript.

Kind regards,

Ayele Mamo Abebe, MSc in pediatric and child health nursing

Academic Editor

PLOS ONE

Journal Requirements:

http://etd.aau.edu.et/bitstream/handle/123456789/21082/Gebremedhin%20Gebreslassie.pdf?isAllowed=y&sequence=1

https://www.hindawi.com/journals/ijrmed/2020/2915628/

https://bmcwomenshealth.biomedcentral.com/articles/10.1186/s12905-021-01218-3

In your revision ensure you cite all your sources (including your own works), and quote or rephrase any duplicated text outside the methods section. Further consideration is dependent on these concerns being addressed.

“N/A”

“NA”

5. We note that Figure 3-6 in your submission contain [map/satellite] images which may be copyrighted. All PLOS content is published under the Creative Commons Attribution License (CC BY 4.0), which means that the manuscript, images, and Supporting Information files will be freely available online, and any third party is permitted to access, download, copy, distribute, and use these materials in any way, even commercially, with proper attribution. For these reasons, we cannot publish previously copyrighted maps or satellite images created using proprietary data, such as Google software (Google Maps, Street View, and Earth). For more information, see our copyright guidelines: http://journals.plos.org/plosone/s/licenses-and-copyright.

 a. You may seek permission from the original copyright holder of Figure 3-6 to publish the content specifically under the CC BY 4.0 license. 

Reviewers' comments:

Reviewer's Responses to Questions

**Comments to the Author**

1. Is the manuscript technically sound, and do the data support the conclusions?

Reviewer #1: Yes

2. Has the statistical analysis been performed appropriately and rigorously? 

Reviewer #1: Yes

3. Have the authors made all data underlying the findings in their manuscript fully available?

Reviewer #1: Yes

4. Is the manuscript presented in an intelligible fashion and written in standard English?

Reviewer #1: Yes

5. Review Comments to the Author

Reviewer #1: Paper Review - PONE-D-23-07027

Spatial distribution, magnitude, and predictors of high fertility status among reproductive age women in Ethiopia: Further analysis of Ethiopia Demography and Health Survey

Title:

Please change “Demography and Health Survey” to “Demographic and Health Survey”. Please indicate they year of the survey for completeness of the title

Cover page

Results – Third line in this section had typo – “reproductive-ag” should read as “reproductive age”

• There is little description regarding spatial variable given the title of these manuscript. In the abstract section, data on “Rural” was present however this is place of residence and can not meet the expectation attached to “spatial distribution”. The authors should include highlights of geographic distribution of the issue across the different regions or other locations to entice readers from the outset and to avoid confusion between simple place of residence and geographic distribution of fertility.

Conclusion – Hot spots are mentioned here but not highlighted to facilitate good reading.

Abstract

Background

Results

• The concern indicated on the summary page which is related to “spatial distribution” applies here. The authors should highlight geographic distribution of fertility.

• Essentially, there seems no difference between the cover page and the abstract section. The authors should differentiate whether the cover page was written in lay language for non-technical readers and if this can be modified accordingly.

Introduction

Method

• Line 115 – “…. The possibility of gave birth…” should read as the possibility of giving birth..

Study Variable

• The authors should reconcile their definition of fertility – the reference in this section talks about the number of children ever born whereas els where they referred to >20weeks of gestation

Results

• Southwestern Oromia (line 215) appeared as one of the high fertility zones in one of the spatial distribution tests. The authors should comment if this was due to differences in the techniques used for analytics and how this should be interpreted. It will be useful to know inter-test harmony in revealing the hot-spots accurately.

Annex

• The title of table 1 – line 450 has typo – the authors should make it sound clearer

6. PLOS authors have the option to publish the peer review history of their article (what does this mean?). If published, this will include your full peer review and any attached files.

Reviewer #1: No

---

## [Author Response · Author response to Decision Letter 0]

10 Jul 2023

Point-by-point response 

Dear reviewer and editors, we would like to extend our deepest appreciation for devoting your time to reviewing our manuscript entitled " Spatial distribution, magnitude, and predictors of high fertility status among reproductive age women in Ethiopia: Further analysis of 2016 Ethiopia Demographic and Health Survey". There has been a revision of this manuscript. The whole structure of the manuscript has been revised. We hope now the manuscript is clear and more acceptable than its previous version. We have tried to present the paper in the proper manner according to your comment on what to supposed to do so. For this, here we have given our responses to each of the concerns you raised. Again, we would like to remind our strongest gratitude for your effort in the improvement of this manuscript, and all the points were addressed in the point-by-point response.

Editor comments: We note that Figure 3-6 in your submission contain [map/satellite] images which may be copyrighted. All PLOS content is published under the Creative Commons Attribution License (CC BY 4.0), which means that the manuscript, images, and Supporting Information files will be freely available online, and any third party is permitted to access, download, copy, distribute, and use these materials in any way, even commercially, with proper attribution. For these reasons, we cannot publish previously copyrighted maps or satellite images created using proprietary data, such as Google software (Google Maps, Street View, and Earth). For more information, see our copyright guidelines: http://journals.plos.org/plosone/s/licenses-and-copyright.

Authors response: Dear, thank you for your concerns, we were writing the source in the caption (legend ) for each figure, kindly see the updated version of the legends. 

Reviewer comments: Please change “Demography and Health Survey” to “Demographic and Health Survey”. Please indicate they year of the survey for completeness of the title

Authors response: Dear reviewer, thank you for your observations and comments, this has been addressed accordingly, kindly see the updated version of the title.

Reviewer comments: Third line in this section had typo – “reproductive-ag” should read as “reproductive age”

Authors response: Thank you for your observations, the typing error has been corrected, kindly see line 44.

Reviewer comments: There is little description regarding spatial variable given the title of these manuscript. In the abstract section, data on “Rural” was present however this is place of residence and can not meet the expectation attached to “spatial distribution”. The authors should include highlights of geographic distribution of the issue across the different regions or other locations to entice readers from the outset and to avoid confusion between simple place of residence and geographic distribution of fertility.

Authors response: Thank you for your comments, this has been modified accordingly, kindly see the updated version of the manuscript.

Reviewer comments: Hot spots are mentioned here but not highlighted to facilitate good reading.

Authors response: Thank you for your observations and comments, this has been modified and addressed, kindly see the updated version of the abstract.

Reviewer comments: The concern indicated on the summary page which is related to “spatial distribution” applies here. The authors should highlight geographic distribution of fertility. Essentially, there seems no difference between the cover page and the abstract section. The authors should differentiate whether the cover page was written in lay language for non-technical readers and if this can be modified accordingly.

Authors response: Dear reviewer, thank you for your observations, this has been addressed accordingly, kindly see the updated version of the manuscript. 

Reviewer comments: Line 115 – “…. The possibility of gave birth…” should read as the possibility of giving birth.

Authors response: Thank you for your observations, the errors were corrected. Kindly see line 116.

Reviewer comments: The authors should reconcile their definition of fertility – the reference in this section talks about the number of children ever born whereas els where they referred to >20weeks of gestation

Authors’ response: thank you for your observations, this has been modified accordingly, kindly see the updated version of the definition and reference.

Reviewer comments: South-western Oromia (line 215) appeared as one of the high fertility zones in one of the spatial distribution tests. The authors should comment if this was due to differences in the techniques used for analytics and how this should be interpreted. It will be useful to know inter-test harmony in revealing the hot-spots accurately

Authors response: Dear reviewer, thank you for your observation and comments, this has been modified accordingly, kindly see the updated version of the manuscript.

Reviewer comments: The title of table 1 – line 450 has typo – the authors should make it sound clear

Authors’ response: Thank you for your observations, this has been corrected, kindly see table 1.

 Thank you

---

## [Editor Report · Decision Letter 1]

21 Aug 2023

Spatial distribution, magnitude, and predictors of high fertility status among reproductive age women in Ethiopia: Further analysis of 2016 Ethiopia Demographic and Health Survey

PONE-D-23-07027R1

Dear Dr. Asmamaw,

We’re pleased to inform you that your manuscript has been judged scientifically suitable for publication and will be formally accepted for publication once it meets all outstanding technical requirements.

Kind regards,

Ayele Mamo Abebe, MSc in pediatric and child health nursing

Academic Editor

PLOS ONE
---

## [Editor Report · Acceptance letter]

29 Aug 2023

PONE-D-23-07027R1 

Spatial distribution, magnitude, and predictors of high fertility status among reproductive age women in Ethiopia: Further analysis of 2016 Ethiopia Demographic and Health Survey 

Dear Dr. Asmamaw:

I'm pleased to inform you that your manuscript has been deemed suitable for publication in PLOS ONE. Congratulations! Your manuscript is now with our production department. 

Kind regards, 

on behalf of

Assistant professor Ayele Mamo Abebe 

Academic Editor

PLOS ONE